# Comparative Effect of Seed Coating and Biopriming of *Bacillus aryabhattai* Z-48 on Seedling Growth, Growth Promotion, and Suppression of Fusarium Wilt Disease of Tomato Plants

**DOI:** 10.3390/microorganisms12040792

**Published:** 2024-04-14

**Authors:** Waheed Akram, Sara Waqar, Sana Hanif, Tehmina Anjum, Zill-e-Huma Aftab, Guihua Li, Basharat Ali, Humaira Rizwana, Ali Hassan, Areeba Rehman, Bareera Munir, Muhammad Umer

**Affiliations:** 1Department of Plant Pathology, Faculty of Agricultural Sciences, University of the Punjab, Lahore 54590, Pakistan; sarawaqar25@gmail.com (S.W.); sana.hanif10@gmail.com (S.H.); tehminaanjum@yahoo.com (T.A.); huma.dpp@pu.edu.pk (Z.-e.-H.A.); alihassanmoon267@gmail.com (A.H.); umerzulfiqar3993@gmail.com (M.U.); 2Guangdong Key Laboratory for New Technology Research of Vegetables/Vegetable Research Institute, Guangdong Academy of Agricultural Sciences, Guangzhou 510640, China; 3Institute of Microbiology and Molecular Genetics, University of the Punjab, Lahore 54590, Pakistan; basharat.ali.mmg@pu.edu.pk; 4Department of Botany and Microbiology, College of Science, King Saud University, Riyadh 11495, Saudi Arabia; hrizwana@ksu.edu.sa; 5College of Earth and Environmental Sciences, University of the Punjab, Lahore 54590, Pakistan; areebarehman453@gmail.com (A.R.); bareeramunir@yahoo.com (B.M.)

**Keywords:** tomato, Fusarium wilt, plant growth-promoting bacteria, Bacillus, seed coating, biopriming

## Abstract

Beneficial plant microbes can enhance the growth and quality of field crops. However, the benefits of microbes using cheap and efficient inoculation methods are still uncommon. Seed coating with biocontrol agents can reduce the amount of inocula along with having the potential for large-scale application. Hence, in this research work, the comparative potential of tomato seed coating and biopriming with *Bacillus aryabhattai* Z-48, harboring multiple plant-beneficial traits, to suppress Fusarium wilt disease along with its beneficial effect on seedling and plant growth promotion was analyzed. Among two bacterial strains, *B. aryabhattai* Z-48 was able to antagonize the mycelial growth of *Fusarium oxysporum* f.sp. *lycopersici* in vitro and its application as a seed coating superiorly benefited seedling traits like the germination percentage, vigor index, and seedling growth index along with a reduced germination time. The seed coating with *B. aryabhattai* Z-48 resulted in significant increases in the shoot length, root length, dry biomass, and total chlorophyll contents when compared with the bioprimed seeds with the same bacterial strain and non-inoculated control plants. The seed coating with *B. aryabhattai* Z-48 significantly reduced the disease index (>60%) compared with the pathogen control during pot trials. Additionally, the seed coating with *B. aryabhattai* Z-48 resulted in a significantly higher production of total phenolics, peroxidase, polyphenol oxidase, and phenylalanine ammonia lyase enzyme in tomato plants. The GC/MS-based non-targeted metabolic profiling indicated that the seed coating with *B. aryabhattai* Z-48 could cause large-scale metabolite perturbations in sugars, sugar alcohols, amino acids, and organic acids to increase the fitness of tomato plants against biotic stress. Our study indicates that a tomato seed coating with *B. aryabhattai* Z-48 can improve tomato growth and suppress Fusarium wilt disease effectively under conventional agricultural systems.

## 1. Introduction

Tomato (*Solanum lycopersicum*) is a widely cultivated vegetable crop in the whole world. The economic importance of tomatoes is undeniable, as they serve as a primary food crop, with a global yield of 181 million tons [1]. Tomato fruit is a dietary source of antioxidants and a rich source of minerals, vitamins, and fiber [2]. The production of tomato crops is prone to different constraints that trigger reduced yield and impaired fruit quality, ultimately reducing the economic benefits for the grower.

One of the main issues with agricultural production around the world is soil-borne diseases, which cause considerable economic damage and negatively affect the quality of several essential crops [3]. Numerous fungal diseases infect tomato plants during their growth cycle. According to an estimate, 40% of the total cost of tomato crops accounts for pest control [4]. The pathogens damage the fruit, stem, and root system of the tomato plant. Fusarium wilt is the most destructive of all diseases. This disease is caused by *Fusarium oxysporum* f.sp. *lycopersici* (Fol). This pathogen is responsible for a ~70% reduction in the crop yield [5]. *F. oxysporum* species are widely distributed around the world in indoor, soil, and marine habitats. The ability of the pathogen to thrive without a host and its long-term persistence in the soil makes it difficult to control the disease with tried-and-true approaches like crop rotation or chemical treatments [6].

Biological control methods have been used for decades to manage agricultural insects and pests. Beneficial microbiomes effectively reprogram the responses of plants that are under stressed conditions and mediate the functioning of the plant micro-ecosystem [7]. The presence of bacterial populations in the rhizosphere plays a significantly vital role in the suppression of plant diseases [8]. The antagonistic function of microbes is essential for disease control as well as spotting possible risks to individuals, the environment, and the emergence of antagonist resistance [9]. *Bacillus* spp. are considered successful biocontrol agents as these can produce antimicrobial chemicals and can elicit plant-induced systematic resistance (ISR) against invading pathogens by producing different secondary metabolites [10,11,12].

The demand for developing novel biological seed coatings is increasing worldwide. Seed coating is usually performed using adhesive agents and filler materials to increase seed performance. Seed coatings are considered an important tool in precision agriculture. Different factors are responsible for the underperformance of microbes applied via conventional inoculation processes like insufficient microbial survival and low inoculum load in the vicinity of emerging seedlings [13]. Tailored seed coatings containing microbial inoculum can effectively alleviate biotic and abiotic stresses and enhance crop growth. Biopriming of seeds with beneficial microbes and seed coating ensures the effective use of small amounts of inoculum at the rhizosphere, guarantees colonization at the germination and early developmental stages, and consequently increases crop production [14]. In addition to a better shelf life and germination efficacy, the microbial coating of seeds can manage soil-borne diseases caused by fungi [14,15,16]. The antagonistic microbes coated onto seeds can safeguard seedlings from different diseases. Secondly, seed coating with microbes is thought to be a cost-efficient technique to deliver microbial inoculants on a large-scale application.

This study was performed to investigate the potential of Bacillus strain/s with multiple plant-beneficial properties to manage Fusarium wilt disease along with the improved performance of tomato seedlings and plant growth. Initially, we determined the antagonistic effect against the Fusarium wilt pathogen in vitro, observed the beneficial effect of bacterial strains on tomato seedling growth, and subsequently used the best-performing bacterial strain to manage Fusarium wilt disease via seed coating and biopriming delivery methods. The ultimate goal of this study was to devise a suitable delivery method of beneficial microbial strain/s to prevent the Fusarium wilt disease of tomato crop.

## 2. Materials and Methods

### 2.1. Tomato Seeds and Bacterial Microbes

The seeds of the Fusarium wilt-susceptible tomato variety “Nema” were used in this study. Two Bacillus plant growth-promoting bacterial strains, “*B. aryabhattai* Z-48 (NCBI Accession: KT027620) and *B. careus* Z-53 (Accession: KT027624)”, were procured from the microbial conservatory of the Institute of Microbiology and Molecular Genetics, University of the Punjab, Lahore, Pakistan. A virulent strain of *Fusarium oxysporum* f.sp. *lycopersici* (Fol) was obtained from the Fungal Biotechnology Laboratory, Department of Plant Pathology, University of the Punjab, Lahore, Pakistan.

### 2.2. Analysis of Antagonism against F. oxysporum

Bacteria were analyzed for antagonism against Fol using a dual-culture technique. Potato Dextrose Agar medium plates were prepared for dual-culture assay. The bacterial isolates were streaked on two sides of the media plate in a straight-line manner. After two days of incubation, a plug (5 mm diameter) of the seven-day-old culture of Fol was placed in the center of the media plate. Each treatment contained three replicates and the experiment was performed twice. Hereafter the incubation period, the percentage inhibition rate was calculated as follows:$$Percent\;Inhibition=\frac{Diameter\;of\;control\;colony-diameter\;of\;treated\;colony}{Diameter\;of\;control\;colony}\times 100$$

### 2.3. Preparation of Treatments

Tomato seeds were surface-sterilized using the standard sodium hypochlorite method as mentioned by Silva, et al. [17]. Seeds were kept under running tap water for one minute. Seeds were dipped in 0.5% sodium hypochlorite for 2 min and 70% ethanol for 2 min. Afterward, the seeds were washed multiple times in distilled sterilized water.

#### 2.3.1. Biopriming of Tomato Seeds

Seeds were bioprimed with both bacteria separately at a culture load of 1 × 10^8^ cfu/mL. To this end, the seeds were dipped in the bacterial suspension for 12 h under continuous shaking conditions. Seeds immersed in sterile distilled water served as the biopriming control.

#### 2.3.2. Coating of Tomato Seeds with Bacterial Strains

The bacterial strains were cultured in a flask containing liquid LB broth media overnight. The cells were separated by centrifugation and aqueous suspension (10^6−7^ c.f.u/mL) was prepared in distilled autoclaved water. Previously, surface-sterilized tomato seeds were coated with both bacterial strains separately by adopting the method of Ehteshamul-Haque, et al. [18] and Chin, et al. [19] using gum arabica solution (5%) and sterilized talc as filler material. Here, coating without any bacterial strain served as the coating process control.

### 2.4. Effect of Seed Coating and Biopriming on Seed Germination and Plant Growth

Coated and bioprimed seeds with bacteria along with the respective controls were sown in the plastic pots containing autoclaved sandy loam soil. Each treatment contained five replicate pots. The experiment was performed in two independent sets. The pots were kept in the greenhouse at 25 ± 3 °C in natural daylight conditions and irrigated with distilled sterilized water. One set of pots was used to analyze the germination and seedling growth. The data were noted to calculate the germination percentage (GP), germination time (GT), germination index (GI), and vigor index (VI) using the below-mentioned formulae [19,20].
GP = (Seeds Germinated)/(Total Seeds) × 100
GT = ∑PF × ∑F
F = the number of seeds newly germinated at the time of X = number of days from sowing.
VI = germination ℅ × mean of seedling length (root + shoot) 
(GI) = ∑Gt/Dt
Gt is the percentage of germination and Dt represents the germination days.

The data regarding plant growth attributes (shoot length, root length, fresh biomass, and dry biomass) were noted 90 days after seed cultivation from the second set of experimental plants. The total chlorophyll and carotenoid contents were estimated by the calorimetric method. The leaf material (1 g) was ground to fine powder in liquid nitrogen and extracted with 80% acetone, centrifuged at 10,000 rpm for 5 min, and the upper clear supernatant was used for chlorophyll estimation. The absorbance values were noted at wavelengths of 645 nm and 663 nm to estimate the total chlorophyll contents using the following equation [21].
Total Chlorophyll (µg/mL) = 20.2(A645) + 8.02 (A663)

### 2.5. Effect of Seed Coating and Biopriming on the Suppression of Fusarium Wilt Disease

#### 2.5.1. Pot Trial

This pot trial was intended to observe the potential of seed coating with bacteria to manage Fusarium wilt disease. The tomato plants were raised from coated, bioprimed, or non-treated seeds in pots filled with sterilized potting media according to experimental design. The aqueous conidial inoculum of Fol was prepared as described in our previous publication [22]. After four weeks of emergence, Fol conidial suspension (50 mL) was injected around the roots of the plants. The experiment consisted of untreated control, pathogen control (Fol), coating with *B. aryabhattai* Z-48 + FOL, biopriming with *B. aryabhattai* Z-48 + Fol, coating process control + Fol, biopriming process control + Fol, fungicide control (Carbendazim @ 15.g/L) + Fol. The plants were incubated under the same conditions as mentioned in Section 2.4. Each treatment consisted of ten biological replicates and the experiment was performed twice.

#### 2.5.2. Data Recorded

The wilt scoring and the disease index were noted 20 days after pathogen application. Wilt scoring was performed according to Hua et al. [23], mentioned in Appendix A. The percentage disease index was noted using the formula of Cachinero, et al. [24]: Percent Disease index = [Σ rating × number of plants rated)/Total number of plants × highest rating] × 100.

Additionally, the shoot length, root length, and dry biomass were analyzed.

#### 2.5.3. Exploration of Mechanisms behind Disease Suppression

The purpose of this experiment was to describe the mechanisms of disease suppression adopting the same methodology as mentioned above with few exceptions. The following treatments were made in this experiment: Untreated control, pathogen control, seed coating with *B. aryabhattai* Z-48 + Fol, seed coating with *B. aryabhattai* Z-48, and seed coating process control. Each treatment contained ten biological replicate tomato plants. Tomato leaves were removed randomly after five days of pathogen application from each treatment and used for the below-mentioned analysis.

##### Quantification of Total Phenolics and Plant Defense-Related Enzymes

The phenolic compounds were extracted from 1 g of leaf material in 80% methanol and were quantified using the Folin–Ciocalteu method. The extracted solution (200 µL) was mixed with 1.4 mL of dist. sterilized water, and 0.1 mL of 50% Folin–Ciocalteu reagent. Afterward, 0.3 mL of 20% (*w*/*v*) sodium carbonate was added to initiate the reaction. The whole reaction mixture was incubated for 2 h in the dark at room temperature. The absorbance was measured at 765. The standard curve was generated by gallic acid and represented in terms of the gallic acid equivalent (GAE) in mg/g of the extract.

For the extraction of defense-related enzymes, the leaf material (1 g) was ground in 2 mL sodium phosphate buffer (0.1 M, pH 7.0). The homogenates were centrifuged for 10 min at 10,000× *g* and the supernatant was used for determining enzyme activities. The activity of peroxidase (POD) was elucidated by Chen, et al. [25]. Briefly, a volume of 50 microliters of enzyme extract was mixed into a phosphate buffer containing 50 microliters of guaiacol (20 mM) as a substrate. The whole mixture was incubated at room temperature for five minutes and the reaction was halted by adding 1 mL of 10% H_2_SO_4_. Changes in absorbance were measured at 470 nm and enzyme activity was expressed as enzyme units g^−1^ FW. One unit of enzyme activity was equal to a 0.01 increase per minute in the corresponding absorbance.

The polyphenol oxidase (PPO) activity was estimated according to the method of Mozzetti, et al. [26]. The mixture contained 0.1 mL of enzyme and enzyme extract (20 µL) in 0.1 M phosphate buffer and 30 mM catechol as a substrate. After incubating for five minutes at room temperature, 1 mL of 10% H_2_SO_4_ was added to terminate the reaction. The resulting change in the reaction mixture was observed by taking the OD at 430 nm. The quantity of enzyme that caused a change of 0.01 in the optical density (OD) value was used to define one unit of polyphenol oxidase activity. The PPO activity was finally expressed as enzyme units g^−1^ FW.

Phenylalanine ammonia lyase (PAL) activity was determined according to the described method by Krishnan et al. [27]. The mixture contained 1.9 mL Tris HCL buffer (0.1 M, Ph 8.5), 0.1 mL enzyme extract, and 1 mL L-phenylalanine (0.015 M) as a substrate. After incubation at room temperature for five minutes, 0.5 mL of 6 M HCl was added to complete the enzymatic reaction. The OD was measured at 280 and enzyme activity was expressed as enzyme units g^−1^ FW. Here, also, one unit of enzyme activity corresponds to the change of 0.01 in the optical density (OD). In the case of all enzymes, the control tubes were made with the addition of dist. water instead of the enzyme extracts.

##### Non-Targeted Metabolome Analysis

Randomly selected leaf samples from five plants of each treatment were pooled to generate one biological replicate. The metabolites were extracted in a solvent (chloroform/methanol/water 4:4:2) in the presence of labeled ribitol as an internal standard [28]. For extraction, plant leaves were pulverized using a pestle and mortar in liquid nitrogen. Fifty mg leaf material was transferred in Eppendorf tubes along with one mL of extraction solution containing ribitol. The solution was sonicated for 15 min in an ultrasonic bath. The solution was centrifuged at 1000 rpm for 5 min to settle down the plant material and suspended particles. The obtained supernatant was further filtered using microfilters (0.2 μm pore diameter) and transferred in new tubes.

The samples were derivatized using MOeX and MSTFA reagents [28]. Briefly, plant extracts were dried using nitrogen gas. An amount of 80 μL of MOeX solution (methoxyl amine hydrochloride @ 20 mg mL^−1^ in pyridine) was mixed with the dried plant extracts and incubated for 2 h at 30 °C. Afterward, MSTFA (80 μL) reagent was added and left for 1.5 h at 37 °C. The derivatized samples were transferred to microinserters for the GC/MS analysis.

The method described by Lisec, et al. [29] was used to analyze the tomato metabolome. Analysis was performed on GCMS-QP2010 (Shimadzu, Kyoto, Japan) fitted with an RT-5 capillary column. Helium was used as the carrier gas. The temperature of the injection and detector were adjusted to 250 °C and 350 °C, respectively. One μL of derivatized plant sample was injected using the splitless mode. Helium gas was used as the carrier gas with a flow rate of 1 mL min^−1^. For GC separation, the temperature was maintained at 50 °C for 1 min, raised to 295 °C @ 10 °C min^−1,^ and was maintained for 8 min. The electron impact was used as an ionization source. Mass spectrometry data were generated in a full-scan mode with a 50–500 m/z range. The NIST library was used to identify the metabolites. Data were analyzed using the Mzmine software package (version 2.5). Obtained values were log10-transformed and normalized to show identical medium peak sizes per sample group.

### 2.6. Statistical Analysis

The data presented are the mean values of the biological replicates from two independent experiments. The data were analyzed statistically by performing ANOVA and DNMRT using the Microsoft Excel (Version 2019) add-on DSAASTAT developed by Anofri (Italy) [30].

## 3. Results

### 3.1. Screening of Antagonistic Bacteria

In the dual-culture analysis, both bacterial strains were screened for antagonistic activity against *F. oxysporum*. The bacterial strain *B. aryabhattai* Z-48 was capable of antagonizing the mycelial growth of *F. oxysporum*. The quantitative analysis revealed 77.25% inhibition of the mycelial growth of F. oxysporum by this strain (Figure 1). The strain B. careus Z-53 was unable to show a zone of inhibition against *B. careus* Z-53; hence, it was excluded from further studies.

### 3.2. Effect of Seed Coating and Biopriming of B. aryabhattai Z-48 on Seedling Growth of Tomato

The seed coating and biopriming with *B. aryabhattai* Z-48 positively affected multiple growth indicators in the tomato seedlings, including the germination, root and shoot length, plant dry matter, and vigor index (Figure 2). Globally, the seed coating with *B. aryabhattai* Z-48 performed superiorly compared to the seed priming treatment as was evident from all the observed parameters. The treatment involving the seed coating with *B. aryabhattai* Z-48 showed the highest germination rate (95.15%) compared to the biopriming (78.26) and non-treated (71.34%) control (Figure 2). The seed coating of *B. aryabhattai* Z-48 significantly reduced the germination time compared with the biopriming and non-treated control (Figure 2). Additionally, the seed coating with *B. aryabhattai* Z-48 significantly increased the germination index (2.2-fold) and vigor index (1.6-fold) compared with the non-treated control treatment. The treatment seed coating process had a non-significant effect on the seedling parameters as compared to the non-treated control plant (Figure 2).

### 3.3. Effect of Seed Coating and Biopriming of B. aryabhattai Z-48 on the Growth of Tomato Plants

For the analysis of tomato plant growth promotion, here, also, we compared two bacterial application methods, i.e., seed coating and seed biopriming under different treatments (Table 1 and Figure 3). The tomato plants attained the maximum shoot length (122 cm) when raised from coated seeds followed by the biopriming application of *B. aryabhattai* Z-48 (104 cm) and non-treated control (83 cm) (Table 1). Similarly, the root length (23.4%), dry biomass per plant (57.6%), and total chlorophyll content (23.9%) were higher in the seed coating treatment compared with the non-treated control (Table 1).

Overall, the seed coating method of the *B. aryabhattai* Z-48 application showed significantly pronounced results in all the growth parameters under analysis compared with the seed biopriming (Table 1). Even though, in the plants raised from biopriming, the shoot length (24.3%), root length (15.2%), dry biomass (37.4%), and total chlorophyll content (18.4%) content were higher compared with the non-treated control plants. The process control treatments (seed coating without bacteria, biopriming with sterilized water) were unable to present a significant effect on the growth-related parameters compared to the non-treated control plants.

### 3.4. Effect of Seed Coating and Biopriming of B. aryabhattai Z-48 on Fusarium Wilt Disease Development and Growth of Tomato Plants

The effects of the seed coating and biopriming with *B. aryabhattai* Z-48 inducing the process controls are shown in Table 2 and Figure 4. Both the seed coating and biopriming application methods significantly reduced the percentage of disease index response in tomato plants against Fusarium wilt disease. The application of the seed coating with *B. aryabhattai* Z-48 superiorly reduced the disease index of Fusarium wilt in tomato plants by up to 63.8% in comparison to the pathogen control (Table 2). Whereas seed biopriming lowered the disease index by up to 48.9% compared with the pathogen control (Table 2).

Compared with the pathogen control (19 cm) plants, the tomato plants raised from coated seeds attained more height (31 cm) as did the plants raised from bioprimed seeds (28 cm). The seed coating with *B. aryabhattai* Z-48 significantly increased the root length (51%) and dry biomass accumulation (36%) and performed significantly superior compared to the seed biopriming application (Table 2).

### 3.5. Exploration of Mechanisms behind Disease Suppression

#### 3.5.1. Effect of Seed Coating on Total Phenolic Compounds and Defense-Related Enzymes

To explore the priming of defense responses in tomato against Fusarium wilt disease, we quantified the total phenolics and defense-related enzymes. As displayed in Figure 5, the levels of the total phenolic content in the pathogen control plants were relatively higher in comparison to the non-treated control plants; however, the total phenolic content was significantly higher in the coated tomato plants challenged with the Fol compared to the pathogen control (>1.5-fold) plant and non-treated control (>2.0-fold) plants (Figure 5).

Similarly, coating with *B. aryabhattai* Z-48 significantly increased the defense-related enzymes compared to the pathogen alone. In the plants obtained from bacterial-coated seeds and challenged with Fol, PO showed the highest increase (2.1-fold) compared to the pathogen control. PPO and PAL lyase increased by up to 1.7- and 2.4-fold in the same scenario (Figure 5).

Plants raised from *B. aryabhattai* Z-48-coated seeds also showed an increase in defense-related enzymes in the absence of the pathogen, but to a lesser extent as compared to the condition where plants were subsequently challenged with the Fol (Figure 5). Fol treatment alone also maintained a higher inducible level of total phenolic and defense-related enzymes in comparison to the non-treated control, but the maximum contents of these were observed in the presence of the bacterial inducer (Figure 5).

#### 3.5.2. Non-Targeted Metabolome Analysis

To further identify defense events in tomato plants in response to Fol and bacterial seed coating, we performed a non-targeted metabolome analysis (Figure 6). After processing the GC/MS data, and performing qualitative and quantities analysis, we identified approximately 40 compounds in tomato plants significantly affected by the application of Fol and *B. aryabhattai* Z-48 in either combination (Figure 7 and Figure 8). To display the differences in the relative quantities of metabolite dynamics during *B. aryabhattai* Z-48 symbiosis and Fol infection, we performed heat map analysis (Figure 8). The total ion chromatograms of tomato metabolites obtained after the treatment applications showed varying abundances of metabolites at different places, indicating reprogramming in different biosynthetic pathways (Figure 8).

After Fol application, the metabolomic responses showed substantial differences compared to the untreated control plants (Figure 6 and Figure 7). Different metabolites were accumulated in restively higher quantities in the Fol-inoculated group in comparison to the non-inoculation control treatment (Figure 7). Whereas the production of numerous primary metabolites including amino acids, sugars, and sugar alcohols was negatively affected by the Fol application compared with the non-treated control plants (Figure 7). The altered metabolites involved in different biosynthetic pathways were summarized on a simplified metabolic map (Figure 7). The seed coating of *B. aryabhattai* Z-48 on the tomato plants in the absence of the pathogen triggered a reprogramming of the primary and secondary metabolism such as increased production of some amino acids, intermediation of TCA cycle, organic acids, sugar, sugar alcohols, etc. (Figure 7 and Figure 8).

As the pathogen alone negatively affected the quantities of metabolites including sugars and some amino acids, the decreased levels were increased under the symbiosis of *B. aryabhattai* Z-48, indicating that the biocontrol agent in the presence of the pathogen cooperatively mediated changes in the host metabolome during infection. In contrast, different metabolites belonging to the phenylpropanoid pathway (caffeic acid, hydroxybenzoic acid, gallic acid, etc.) were increased when the tomato plants were challenged by the pathogen alone compared with the non-treated control (Figure 7 and Figure 8). The levels of these phenylpropanoids were further increased in the plants raised from seeds coated with *B. aryabhattai* Z-48 and subsequently challenged with the pathogen (Figure 7 and Figure 8).

## 4. Discussion

Tomato, an important vegetable crop, can be severely affected by biotic stresses including Fusarium wilt disease caused by *F. oxysporum*. This study focused on the delivery of bacterial microbes with multiple beneficial properties using biopriming and seed coating methods. This study further investigated the symbiosis of bacterial microbes on seed germination, plant growth, and disease suppression. The mechanisms behind the mentioned beneficial properties were described using morphological, physiological, and metabolite changes in tomato plants.

Two rhizospheric bacterial strains, “*B. aryabhattai* Z-48 and *B. careus* Z-53”, with plant-documented plant growth-promotion properties were initially screened to antagonize the radial growth of F. oxysporum using a dual-culture technique. We discovered that *B. aryabhattai* Z-48 significantly antagonized fungal phytopathogen under in vitro plate assay, showing >70% fungal growth inhibition. Different species of Bacillus synthesize hydrolytic enzymes and can beneficially modify the environment surrounding plant roots [31]. These can also produce cell wall-degrading enzymes and several secondary metabolites capable of limiting the mycelial growth of fungi [32]. The findings of the in vitro experiment suggested that *B. aryabhattai* Z-48 could be an effective biological control agent against Fusarium wilt disease due to having direct antagonism. Hence, it was selected for further downstream studies. Previous studies depict that crops, like vegetables, cultivated in greenhouses serve as excellent sources for displaying the positive potential of beneficial microbes [33]. Hence, it can be postulated that the plant-beneficial traits of *B. aryabhattai* Z-48 can be effectively replicated in tomato plants under field conditions.

Previous studies mostly reported PGPR inoculation via the conventional direct application method in the soil. We adopted two delivery methods (seed biopriming and seed coating) for establishing symbiosis among tomato plants and *B. aryabhattai* Z-48, which was selected based on the in vitro studies.

Coating with *B. aryabhattai* Z-48 superiorly benefited the tomato seedling growth and showed the maximum growth of tomato plants among different treatments, while seed biopriming with *B. aryabhattai* Z-48 brought an inferior performance in the seedling growth and different plant growth parameters. The same has been observed in different previous studies using *Bacillus* spp. [34,35,36]. This could be because seed coating enhances microorganism delivery, fostering root–plant interaction during germination. We observed a significantly improved germination rate, germination index, and vigor index. The presence of beneficial bacteria in the seed coating can trigger different physiological processes within the seed that ultimately boost the plant growth levels [15]. Additionally, the proliferation of beneficial bacteria around the seeds protects from pathogens, allowing the seedling to withstand adverse conditions [37]. It was also observed that plants raised from coated seeds comprised a significantly increased shoot length, root length, and dry biomass as these beneficial bacterial microbes can produce plant growth regulators, such as auxins, cytokinins, abscisic acid, and gibberellins [38].

The presence of rhizospheric bacteria capable of antagonizing plant pathogens in seed coatings can offer a promising strategy for controlling soil-borne diseases along with plant growth promotion. During pot trials, seeds coated with *B. aryabhattai* Z-48 significantly reduced the disease index of Fusarium wilt on tomato plants, leading to improved plant growth and disease resistance. Seed coating additionally influenced growth indicators, including the shoot length, root length, and plant dry biomass, under pathogen stress compared to the pathogen control plants. Different previous studies have also documented the multiple beneficial properties of rhizospheric bacteria including plant growth promotion and disease suppression. *B. amyloliquefaciens* strain B9601-Y2 suppressed Bipolaris leaf spots in maize plants along with an increased plant growth [39].

Plants possess phenolic compounds that play a major role in hindering invading pathogens. Phenolic compounds are pivotal to local or systemic resistance in plants. Phenolic compounds also play a major role in induced systemic resistance. These toxic compounds can work effectively against soil-borne pathogens [40]. Our findings also indicate the role of the seed coating with *B. aryabhattai* Z-48 in the significantly increased production of total phenolic and defense-related enzymes in tomato plants. This mechanism of up-regulated defense responses is worthwhile for the management of soil-borne plant diseases that are otherwise difficult to control using conventional agricultural practices. This also helps to reduce pathogen colonization inside the plant body where direct contact between the biocontrol agent and plant pathogen is not possible [41]. Previous studies have indicated the role of Bacillus bacteria in the increased biosynthesis/activities of total phenolics and enzymes involved in the phenylpropanoid pathway, responsible for flavonoids, polyphenols, and lignin production [42]. In this study, we observed increased activities of peroxidase (PO), polyphenol oxidase (PPO), and phenylalanine ammonia lyase (PAL) enzymes in tomato plants raised by seeds coated with *B. aryabhattai* Z-48. It can be speculated that *B. aryabhattai* Z-48 contains conserved molecular signatures. Upon symbiosis, plants perceive these signatures, which in turn induces increased defense responses and/or activation of defense-related enzymes.

We performed a non-targeted comprehensive analysis of changes in the metabolic profile of coated tomato plants and responses to the Fol. The results showed that the Fusarium wilt pathogen could cause numerous metabolic changes, e.g., reduced production of different sugars, organic acids, and amino acids. We constructed a simplified pathway analysis displaying the metabolomic data. This demonstrated that amelioration/s was seen in the metabolic levels in the bacterial-inoculated tomato plants that were subsequently challenged with Fol. Symbiosis of *B. aryabhattai* Z-48 significantly increased the production of different carbohydrates, sugar alcohols, amino acids, and sugar alcohols. Noteworthy, the pathogen alone significantly increased the production of some phenolic acids in the tomato plants compared to the non-treated control plants. Whereas the seed coating with *B. aryabhattai* Z-48 further increased the production of these defense-related compounds. Previous studies have reported pronounced changes in the metabolomic profile of plants under the influence of pathogens and beneficial microbes. In a study, tomato plants were primed by four different plant growth-promoting bacterial strains. These strains induced considerable reprogramming in defense-related metabolic pathways. It was characterized by changes in the levels of hydroxycinnamates, flavonoids, and alkaloids [43]. Similarly, *Bacillus subtilis* and *Paenibacillus alvei* induced metabolic reconfigurations in wheat plants against rust disease caused by *Puccinia striiformis* [44]. The metabolic changes observed in our study display a preconditioned state with induced defense priming that renders resistance against the Fusarium wilt pathogen. These results will contribute to unraveling the biochemical responses that define the priming phenomenon mediated by the beneficial microbial inside host plants.

## 5. Conclusions

Food insecurity is a global concern driven by population growth and increased food consumption. Our study describes for the very first time the application of *B. aryabhattai* via seed coating to suppress Fusarium wilt disease along with better growth and seedling establishment. This study highlights the potential of *B. aryabhattai* strain Z-48 to be used in conventional agriculture systems to enhance crop productivity along with the pretension against a destructive soil-borne disease.

## Figures and Tables

**Figure 1 microorganisms-12-00792-f001:**
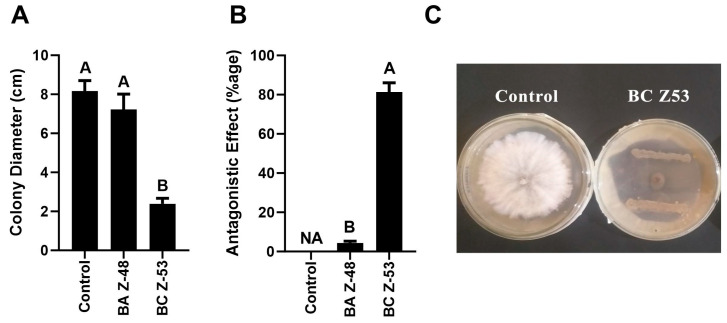
Analysis of the antagonistic potential of rhizospheric bacteria. (**A,B**). Inhibition percentage of antagonistic bacterial strains. (**C**). Qualitative screening of antagonistic bacterial strains. Vertical bars represent standard error. Different capital letters show significant differences among treatments at *p* = 0.05. BA = *B. aryabhattai* Z-48; BC = *B. cereus*.

**Figure 2 microorganisms-12-00792-f002:**
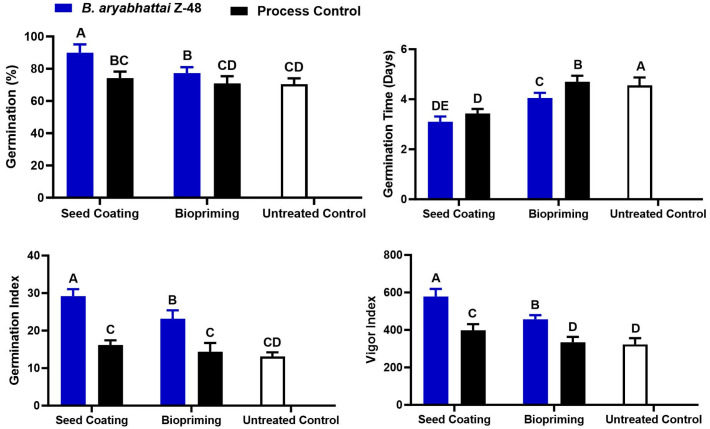
Analysis of the effect of coating on seedling quality of tomato. Vertical bars represent standard error. Different capital letters show significant differences among treatments at *p* = 0.05. Process control for seed coating represents the coating of seeds with coating material without bacteria and for biopriming represents seeds dipped in water only without bacteria.

**Figure 3 microorganisms-12-00792-f003:**
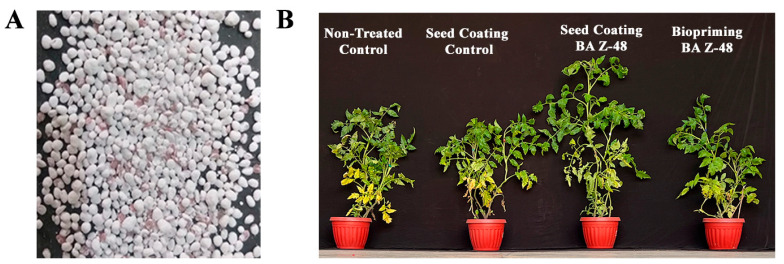
Effect of seed coating and biopriming on the growth of tomato plants. (**A**) = Coated seeds of tomato plants. (**B**) = Mature tomato plants raised from coated and bioprimed seeds of tomato by *Bacillus aryabhattai* Z-48.

**Figure 4 microorganisms-12-00792-f004:**
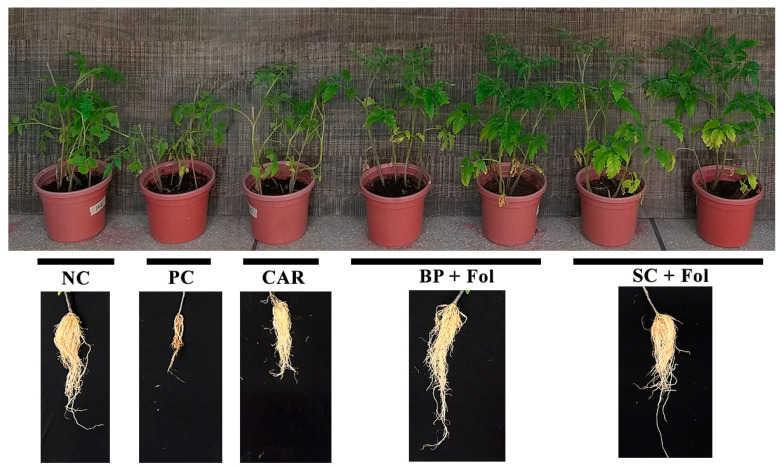
The potential of combined application of bacterial consortia and chemoattractant to manage Fusarium wilt of tomato. NC = non-treated control; PC = pathogen control; CAR = carbendazim control; BP = biopriming with *B. aryabhattai* Z-48; SC = seed coating with *B. aryabhattai* Z-48; and Fol = *F. oxysporum* f.sp. *lycopersici*.

**Figure 5 microorganisms-12-00792-f005:**
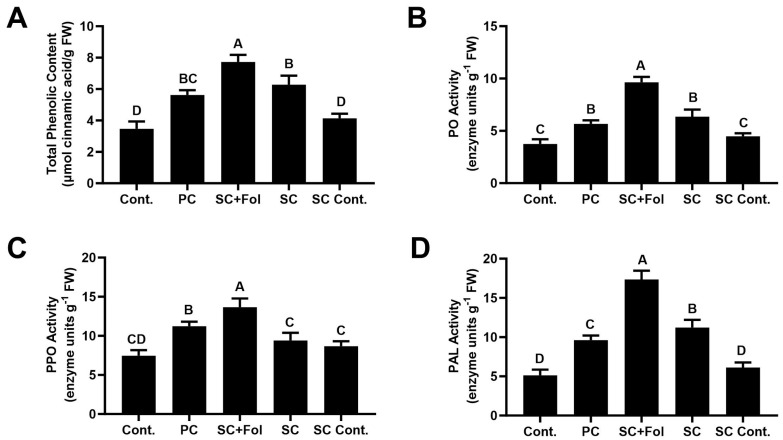
The effect of different treatments of bacterial seed coating on the defense mechanisms in tomato plants. Vertical bars represent standard error. Different capital letters show significant differences among treatments at *p* = 0.05. Cont. = non-treated control plants. (**A**) Quantification of total Phenolic Content; (**B**) Peroxidase Activity; (**C**) Polyphenol oxidase Activity; (**D**) Phenyl alanine ammonia lyase Activity. PC = pathogen control. SC = seed coated with *B. aryabhattai* Z-48. Fol = *F. oxysporum* f.sp. *lycopsrsici*. SC Cont. = seed coating without *B. aryabhattai* Z-48 to serve as seed coating process control.

**Figure 6 microorganisms-12-00792-f006:**
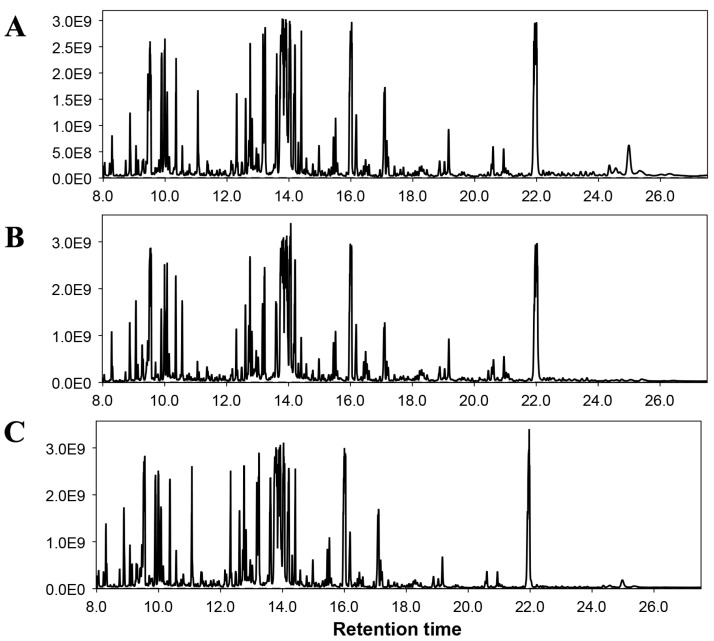
Representative GC/MS chromatograms of tomato plants. (**A**) = Chromatogram of untreated control plants. (**B**) = Chromatogram of pathogen control, and (**C**) = chromatogram of tomato plants raised by seeds coated with *B. aryabhattai* Z-48 and subsequently challenged with *F. oxysporum*.

**Figure 7 microorganisms-12-00792-f007:**
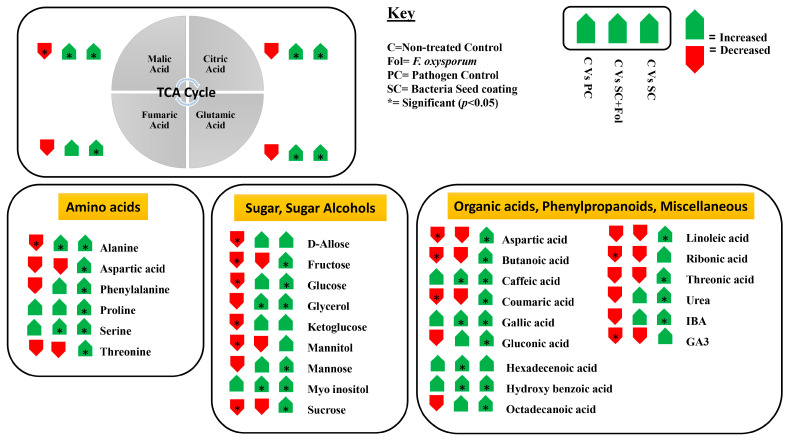
Metabolic pathway model of changes in tomato metabolome mediated by seed coating with *B. aryabhattai* Z-48 and Fusarium wilt pathogen. The green arrow shows increased production of metabolite whereas the red shows decreased production. The data presented here were derived from non-targeted GC/MS analysis of tomato plants raised under different treatments. Quantitative analysis was performed among different treatments and compared to the non-treated control plants. The quantities of each metabolite among replicates were averaged. Five biological replicates were used to derive the data. (*) = Significant difference (*p* > 0.05).

**Figure 8 microorganisms-12-00792-f008:**
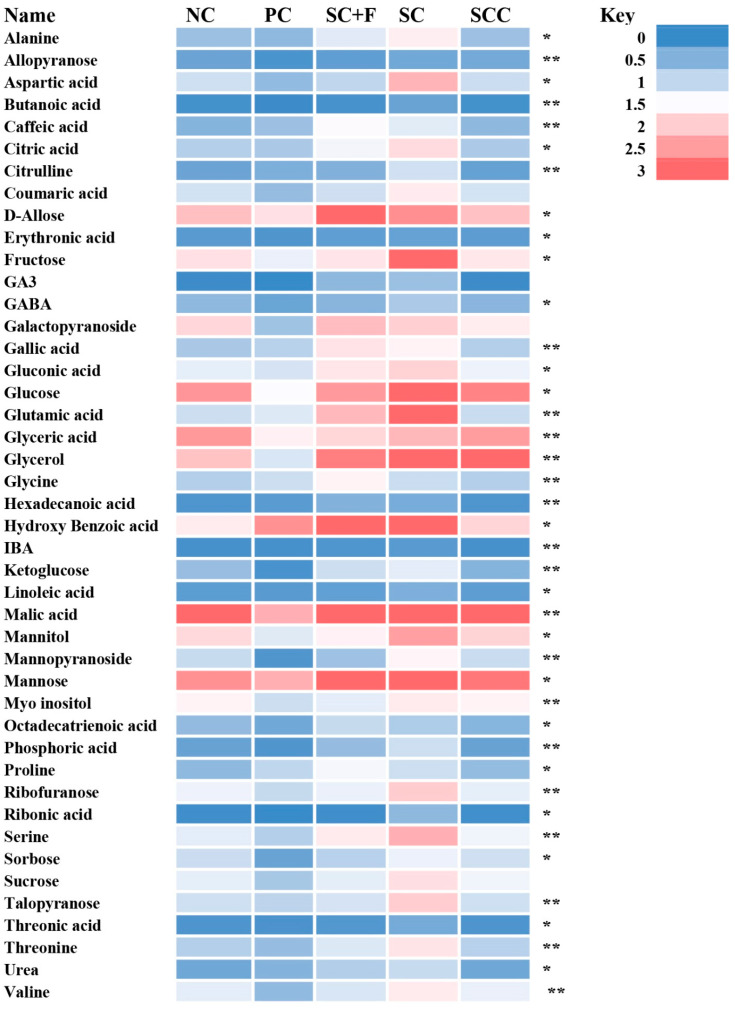
Heatmap of relative metabolite levels in the leaves of tomato plants raised from coated seeds with *B. aryabhattai* Z-48 and exposed to Fusarium wilt pathogen. NC = non-treated control; PC = pathogen control; SC = seed coated with *B. aryabhattai* Z-48; F = Fol; and SCC = seed coating control. * = *p* > 0.05; ** = *p* > 0.01.

**Table 1 microorganisms-12-00792-t001:** Effect of seed coating and biopriming on the growth of tomato plants.

Treatments		Shoot Length (cm)	Root Length (cm)	Dry Biomass (g)	Total chl. (mg/g FW)
Seed coating	BA Z-48	122 ± 11.93 ^A^	34.57 ± 01.21 ^A^	93.5 ± 05.47 ^A^	1.71 ± 0.01 ^A^
	Process control	86.9 ± 07.31 ^C^	28.96 ± 02.17 ^C^	66.7 ± 03.12 ^C^	1.23 ± 0.03 ^C^
Biopriming	BA Z-48	104 ± 08.06 ^B^	32.58 ± 01.82 ^AB^	87.3 ± 06.12 ^B^	1.64 ± 0.05 ^AB^
	Process control	81.7 ± 06.13 ^CD^	29.42 ± 01.02 ^C^	61.2 ± 05.12 ^C^	1.32 ± 0.02 ^C^
Non-treated control		83.2 ± 08.22 ^CD^	27.46 ± 01.74 ^CD^	59.1 ± 07.09 ^CD^	1.38 ± 0.01 ^CD^

Values show mean ± standard error. Different capital letters show significant differences among treatments at *p* = 0.05. BA Z-48 = *B. aryabhattai* Z-48. Seed coating process control represents seed coating without bacteria. Whereas biopriming process control represents priming with water only.

**Table 2 microorganisms-12-00792-t002:** Effect of different treatments on suppression of Fusarium wilt disease and growth attributes of tomato.

Treatments		Disease Index (%age)	Shoot Length (cm)	Root Length (g)	Dry Biomass (g)
Seed coating	BA Z-48	23.2 ± 01.86 ^E^	31.8 ± 02.25 ^A^	16.7 ± 01.29 ^A^	8.54 ± 0.65 ^A^
	Process control	69.6 ± 05.42 ^AB^	22.9 ± 01.86 ^C–E^	07.4 ± 09.16 ^C–E^	5.92 ± 0.31 ^CD^
Biopriming	BA Z-48	35.4 ± 02.27 ^CD^	28.4 ± 01.46 ^AB^	14.2 ± 01.47 ^B^	7.66 ± 0.92 ^AB^
	Process control	75.7 ± 04.94 ^A^	20.0 ± 01.43 ^DE^	08.6 ± 00.94 ^CD^	5.34 ± 0.44 ^CD^
Carbendazim control		41.9 ± 02.67 ^C^	23.8 ± 00.96 ^CD^	10.9 ± 07.09 ^C^	6.94 ± 0.52 ^BC^
Pathogen control		73.2 ± 04.52 ^A^	19.3 ± 01.07 ^EF^	08.31 ± 00.64 ^CD^	5.31 ± 0.37 ^CD^
Non-treated control		ND	25.9 ± 00.86 ^BC^	13.96 ± 01.27 ^B^	6.18 ± 0.42 ^BC^

Values show mean ± standard error. Different capital letters show significant differences among treatments at *p* = 0.05. BA Z-48 = *B. aryabhattai* Z-48. Seed coating process control represents seed coating without bacteria. Whereas biopriming process control represents priming with water only.

## Data Availability

The data supporting the findings of this study are available within the article and Appendix A.

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
