# Peer review of "Comparative Effect of Seed Coating and Biopriming of Bacillus aryabhattai Z-48 on Seedling Growth, Growth Promotion, and Suppression of Fusarium Wilt Disease of Tomato Plants"

_microorganisms, 2024, doi:10.3390/microorganisms12040792_

Round 1

Reviewer 1 Report

Comments and Suggestions for Authors

Dear authors,

 I have read the manuscript carefully and I must note that it contains valuable data. At the same time, there are some shortcomings, unclear aspects or incompletely described methodological elements that should be reviewed carefully before publication.

The Introduction section gives an overview and frames the research objective in the more general trend of studies regarding the beneficial influence of bacteria associated with plants.

Materials and Methods section - The description of methods for estimating defense-related enzymes is incomplete. The authors must mention the essential technical elements so that the experiments can be reproduced.The essential elements of the experimental protocol are missing in the subsections " 2.5.2.1. Quantification of total phenolics and plant defense-related enzymes” (lines 186-202) as well as    "2.5.2.2. Non-targeted metabolome analysis " (lines 204-206; 212-213; 215).

Lines 193-194. Please correct and replace the word "elucidated" with the more correct version "described". And add details of the essential elements of the method in such a way that other researchers can reproduce your experiment.

Regarding the experimental protocol - Enzyme activity was estimated based on the determination of the amount of protein. Many types of proteins can be found in the supernatant. How do the authors know which of those proteins have the activity of the three enzymes? An enzyme, even though it is a protein, has a very specific substrate that it catalyzes and leads to the formation of a final product. What are the substrates used in the experiments to identify the enzymes mentioned in the article and their reaction product?

Results section. Figure 5, lines 186-202.

 In the text we found references to peroxidase, polyphenol oxidase and phenylalnine ammonia oxidase, but in the caption of Figure 5 the respective enzymes cannot be distinguished. Axis Oy contains reference only to proteins. What is the proof that the authors really detected the respective enzymes and not just the protein content of the supernatant?

What is the significance of total phenols in the defense mechanisms of plants? Insert additional information from the literature to justify the use of this parameter in experiments.

In a broader perspective, can the results of experiments carried out in limited and controlled conditions be reproduced in natural conditions? This is because, very likely, the bacterial strain used in biopriming and coating will interact and compete with a multitude of microbial species in external conditions very different from those in the experiments.

I ask the authors to estimate, at least theoretically, which of the experimental results could be observed in plants grown in natural conditions.

Figure 7, lines 37-374. The diagram presents mixed data, difficult for readers to understand. I ask the authors to clearly separate the control from the experimental variants (and not C vs PC; C vs SC+F; C vs SC; C vs SCC).

In Conclusions section, authors should emphasize what is new in their article in relation to the research of other authors in the field.

With best regards!

Comments on the Quality of English Language

In general, the English is good, but the text contains some typographical errors that need to be corrected.

Author Response

Thanks to the respected reviewers for the valuable comments. Definitely, it will improve the quality manuscript. Relevant material has been added and changes incorporated as advised. The text color of the amended material has been changed to red for the ease of respected viewers.

Comment:  Materials and Methods section - The description of methods for estimating defense-related enzymes is incomplete. The authors must mention the essential technical elements so that the experiments can be reproduced. The essential elements of the experimental protocol are missing in the subsections " 2.5.2.1. Quantification of total phenolics and plant defense-related enzymes” (lines 186-202) as well as    "2.5.2.2. Non-targeted metabolome analysis " (lines 204-206; 212-213; 215).

Response:

The experimental protocol has been substantially revised under section (2.5.2). Detailed methods of quantification of phenolics and defense-related enzymes have been added (line 182-216).

Comment:  Lines 193-194. Please correct and replace the word "elucidated" with the more correct version "described". And add details of the essential elements of the method in such a way that other researchers can reproduce your experiment.

Response:

The word “elucidated” has been replaced by the words “describe” and “exploration”. (line 174, 175, 329, 331).

Comment:  Regarding the experimental protocol - Enzyme activity was estimated based on the determination of the amount of protein. Many types of proteins can be found in the supernatant. How do the authors know which of those proteins have the activity of the three enzymes? An enzyme, even though it is a protein, has a very specific substrate that it catalyzes and leads to the formation of a final product. What are the substrates used in the experiments to identify the enzymes mentioned in the article and their reaction product?

Response:

Specific substrates were used for each enzymes. These details have been added in this revised manuscript. (Line 182-216)

Comment:  Results sectionFigure 5, lines 186-202.

 In the text we found references to peroxidase, polyphenol oxidase and phenylalnine ammonia oxidase, but in the caption of Figure 5 the respective enzymes cannot be distinguished. Axis Oy contains reference only to proteins. What is the proof that the authors really detected the respective enzymes and not just the protein content of the supernatant?

Response:

Figure 5 and the caption has been revised adding the information about each enzyme. (line 344-348). Similarly, axis of graphs have been modified to convey the name of the enzyme.

Comment:  What is the significance of total phenols in the defense mechanisms of plants? Insert additional information from the literature to justify the use of this parameter in experiments.

Response:

The significance of total phenols has been added in the discussion section. (line 458-461)

Comments: In a broader perspective, can the results of experiments carried out in limited and controlled conditions be reproduced in natural conditions? This is because, very likely, the bacterial strain used in biopriming and coating will interact and compete with a multitude of microbial species in external conditions very different from those in the experiments.

I ask the authors to estimate, at least theoretically, which of the experimental results could be observed in plants grown in natural conditions.

Response:

Advised material has been added in the discussion section. (line 425-428)

Comments: Figure 7, lines 37-374. The diagram presents mixed data, difficult for readers to understand. I ask the authors to clearly separate the control from the experimental variants (and not C vs PC; C vs SC+F; C vs SC; C vs SCC).

Response: As per suggestions data in Figure 8 has been revised and control is separated from the rest of the treatment. More information has been added in the caption of the figure 7. As Figure 7 provides the overview of the whole metabolomics analysis. It is deemed necessary to provide their comparison with the control and this is widely accepted method. So, this may kindly be accepted.

Comments: In Conclusions section, authors should emphasize what is new in their article in relation to the research of other authors in the field.

Response:

Conclusion section has been revised and the novelty statement has been mentioned in the conclusion section. (line 499-504).

Reviewer 2 Report

Comments and Suggestions for Authors

Results presenting by Authors are interesting with good potential link to practice.

The introduction gives the reader sufficient background before results analyses;

Manuscript is generally very well written, especially discussion part is without often unnecessary redundancy.

Moreover, in my opinion from the discussion emerges the importance of conducting and presented research.

In my opinion the manuscript should be published, but after some inaccuracies corrections:

·       I suggest to rethink better organization in part “Quantification of total phenolics and plant defense-related enzymes” – methodology is clear, but the results presentation rather not; - Please rebuild Figure 5 this part did not correspond with methodology;

·       In my opinion only Figure 7 (first figure 7:) need to be enlarged and quite more explained in figure caption, because some details are lost for the readers.

·       Unfortunately I find two figures 7- please correct mistake;- but these figure is very informative and well-constructed.

·       Conclusions are rather a some part of future prospects coming from results and gives potential to use presenting methods in practice.

Author Response

Thanks to the respected reviewers for the valuable comments. Definitely, it will improve the quality manuscript. Relevant material has been added and changes incorporated as advised. The text color of the amended material has been changed to red for the ease of respected viewers.

Comment: I suggest to rethink better organization in part “Quantification of total phenolics and plant defense-related enzymes” – methodology is clear, but the results presentation rather not; - Please rebuild Figure 5 this part did not correspond with methodology;

Response: Figure 5 has been improved incorporating the necessary information.

Comment: In my opinion only Figure 7 (first figure 7:) need to be enlarged and quite more explained in figure caption, because some details are lost for the readers.

Comment: Unfortunately I find two figures 7- please correct mistake;- but these figure is very informative and well-constructed.

Response: The figure caption has been revised adding necessary details (lines 397-403). Digit “7” has been replaced with “8”.  Figure has been increased in the size.

Comment: Conclusions are rather a some part of future prospects coming from results and gives potential to use presenting methods in practice.

Response: Conclusion section has been revised adding the relevant information. (line 501-506).

Round 2

Reviewer 1 Report

Comments and Suggestions for Authors

 Dear authors,

 Thank you for the seriousness and speed with which you made changes to the manuscript according to our recommendations. The quality and clarity of the text have been obviously improved and the manuscript is publishable.

With best regards!